# Interoceptive Processing in Functional Gastrointestinal Disorders

**DOI:** 10.3390/ijms25147633

**Published:** 2024-07-11

**Authors:** Katerina Karaivazoglou, Ioanna Aggeletopoulou, Christos Triantos

**Affiliations:** 1Department of Psychiatry, University Hospital of Patras, 26504 Patras, Greece; karaivaz@hotmail.com; 2Division of Gastroenterology, Department of Internal Medicine, University Hospital of Patras, 26504 Patras, Greece; iaggel@upatras.gr

**Keywords:** functional gastrointestinal disorders, interoception, insula, gut–brain axis

## Abstract

Functional gastrointestinal disorders (FGIDs) are characterized by chronic gastrointestinal symptoms in the absence of overt pathology and affect a significant percentage of the worldwide population. They are commonly accompanied by co-morbid psychiatric symptomatology and are associated with significant suffering and great healthcare services utilization. There is growing evidence that dysregulation of the gut–brain axis and disturbances in the processing of afferent interoceptive signals lie at the heart of these disorders. In this context, the aim of the current review was to detect and critically review original articles focusing on the role of interoception in the pathophysiology of FGIDs. Our search yielded 38 relevant studies. FGID patients displayed increased visceral sensitivity, enhanced attention to gastrointestinal interoceptive cues, and greater emotional arousal when coping with gut-derived sensations. Neuroimaging studies have shown significant structural and functional changes in regions of the interoceptive network, while molecular and genetic studies have revealed significant associations between interoceptive signaling and deficits in excitatory neurotransmission, altered endocrine and immune physiological pathways, and aberrant expression of transient receptor potential channel genes. Finally, there were emerging data suggesting that interoception-based interventions may reduce physical symptoms and improve quality of life and should be integrated into FGID clinical management practices.

## 1. Introduction 

Functional gastrointestinal disorders (FGIDs) constitute a group of highly prevalent conditions characterized by chronic gastrointestinal symptoms in the absence of overt pathology on conventional testing. According to the current classification system (Rome IV), this diagnostic category includes 33 adult and 20 pediatric disorders, with the most common being irritable bowel syndrome (IBS) and functional dyspepsia (FD). FGIDs are associated with increased psychiatric co-morbidity, impaired quality of life (QoL), and great healthcare utilization, thus imposing a huge burden on individual patients and healthcare systems [1]. Although termed functional, recent research has revealed that physiological alterations, including changes in gastrointestinal motility, disturbances of the brain–body communication, and aberrant processing of sensory stimuli, may be implicated in FGID pathophysiology [2]. More specifically, there is a growing body of evidence linking gastrointestinal symptomatology with an atypical pattern of processing of interoceptive stimuli originating from the gastrointestinal tract, while accumulating neuroimaging data suggest that there are significant differences in the activation and connectivity patterns of brain regions involved in interoception in patients with FGIDs compared to other chronic gastrointestinal diseases and healthy individuals [3,4]. 

The term interoception refers to the detection, transmission, and central processing of internal bodily sensations, including heart rate, respiration, fatigue, thirst, hunger, satiety, and visceral pain/discomfort. It is a multidimensional construct that encompasses bottom-up (afferent signaling) and top-down (central processing, integration of other modalities’ sensory input, and neural and mental representation of internal bodily signals) pathways [5,6]. It comprises three distinct processes: interoceptive accuracy, interoceptive sensibility, and interoceptive awareness. Interoceptive accuracy is defined as the ability to detect and track internal bodily sensations, which can be measured by objective performance tests. Interoceptive sensibility refers to the subjective belief regarding one’s interoceptive abilities and the degree of one’s engagement with interoceptive signals; it is evaluated with the use of self-administered questionnaires. Finally, interoceptive awareness is the metacognitive awareness of interoceptive abilities and is reflected by the correspondence between objective interoceptive accuracy and subjective reports [7,8]. Interoceptive signaling constitutes a critical component of brain–body communication and contributes to the body’s homeostatic control, emotional processing, and the formation of a conscious unitary sense of self and others. Given that interoceptive information plays a large part in the shaping of emotional experience, individual emotional attitudes may reflect differences in interoceptive sensitivity and processing [6]. Interoception is mediated by a complex and widespread neural circuit of afferent and efferent pathways and is regulated by a plethora of cortical and sub-cortical structures, including the brain stem, the insula, the anterior cingulate, the amygdala, the hippocampus, the basal ganglia, and the somatosensory cortex [6,9] (Figure 1).

In this context, the scope of the current article is to critically review original studies investigating the role of interoceptive processing in the pathophysiology of functional gastrointestinal diseases. To do so, we performed a thorough literature search using a variety of relevant terms in any possible combination in an attempt to extract as much information as possible (Table 1). 

In addition, we performed a manual search using all references of the selected articles to detect additional relevant literature. Our search exclusively included articles written in English and published during the last decade (2014–2024). Studies in non-human subjects were excluded from the current review. The final search was conducted at the beginning of June 2024. Table 2 includes all studies which were considered relevant and whose findings were reviewed for the current article. 

## 2. Studies Characteristics

Our search retrieved 38 original research articles that were relevant (Table 2). Thirty of these studies included irritable bowel syndrome (IBS) patients [10,11,12,13,14,15,19,20,21,22,23,24,26,27,28,29,31,32,34,36,37,38,39,40,41,43,44,45,46,47], five studies included functional constipation (FC) patients [17,18,25,30,42], one study enrolled functional dyspepsia (FD) patients [16], one study recruited functional abdominal pain patients [35], and one study included a mixed group of abdominal pain-related functional gastrointestinal disorder patients [33]. Twenty-seven (27) of these studies had a cross-sectional design [11,12,13,14,16,17,18,20,22,24,25,26,27,29,30,31,32,34,36,37,38,41,42,44,45,46,47], eight studies were clinical trials [10,15,19,23,33,35,40,43] (five randomized controlled-RCT [15,23,33,40,43]) assessing the effect of interoceptive-altering manipulations or the exogenous administration of 5-hydroxytryptophan [43] or ebastine [40] on patients’ symptomatology, one (1) was a single-blind controlled study [39] assessing the effect of exogenous corticotropin-releasing hormone (CRH) administration on brain region activation during colorectal distension, and two (2) were prospective controlled studies assessing vagal tone and rectal sensitivity during colorectal distension [28] and rectal sensitivity after small bowel lipid infusion [21], respectively. The majority of studies included control groups, either healthy individuals or patients with inflammatory bowel disease (IBD) or functional nonretentive fecal incontinence (FNRFI); however, four studies did not have controls in their sample [13,19,26,35]. In 21 studies, interoceptive processing was assessed with the use of neuroimaging techniques [12,16,17,18,20,25,26,27,30,31,32,34,36,37,38,39,42,44,45,46,47]. In four studies, interoception was assessed with an objective performance test, namely the two-stage water load test [11] and the rectal sensitivity test [21,22,43]. In four studies, interoceptive sensibility was evaluated with the use of self-administered questionnaires such as the Interoceptive Sensitivity and Attention Questionnaire (ISAQ) [13], a Visual Analogue Scale of abdominal pain severity [14], the Difficulties in Interoceptive Abilities (DIA) subscale of the Toronto Alexithymia Scale (TAS) [24], and the Visceral Pain Questionnaire [41]. In one investigation, interoceptive signaling was evaluated using Ca^2+^ imaging of human sensory neurons [29], while in another, a combination of a rectal sensitivity test and Holter electrocardiography to calculate heart rate variability as an index of parasympathetic tone was used [28]. Among neuroimaging studies, in twelve studies, the researchers used fMRI [16,17,18,26,30,32,34,37,38,42,45,46]; in seven studies, structural MRI was used [12,20,25,31,36,44,47], and in the remaining two studies, an MRS [27] or PET scan [39] was performed.

## 3. Irritable Bowel Syndrome 

Irritable bowel syndrome (IBS) is the most prevalent FGID and is characterized by altered bowel habits accompanied by abdominal discomfort or pain. Its pathophysiology is multifactorial and involves microbiota dysbiosis, disturbances of the gut–brain axis, heightened visceral sensitivity, and low-grade intestinal inflammation [48]. In a recent investigation [11], patients with IBS exhibited increased gastric sensitivity compared to patients with IBD, and higher gastric sensitivity was associated with worse emotional quality of life (QoL) and impaired social functioning. Likewise, in another study [22], patients with IBS reported increased visceral sensitivity and a decreased visceral pain threshold compared to healthy controls. These findings suggest that in IBS, there is an augmented afferent interoceptive signaling, which probably affects patients’ emotional and behavioral responses to gastrointestinal stimuli. Bogaerts et al. (2022) [13], assessing self-reported interoceptive sensibility, showed that patients with IBS report increased sensitivity to neutral bodily sensations and increased attention to unpleasant bodily sensations compared to healthy controls. In addition, another study recently revealed that patients with IBS display increased visceral pain and heightened difficulties in interoceptive abilities compared to healthy controls and ulcerative colitis patients; the authors linked these differences with alexithymic traits and difficulties in the discrimination between emotions and bodily sensations [24].

Interoceptive information reaches higher-order sub-cortical and cortical structures in the brain through variable ascending pathways, namely humoral, lymphatic, and neural. The major neural interoceptive afferent pathways include the vagus, the other cranial nerves, and the spinal afferents [49]. Patients with IBS had a blunted sympathovagal balance response to rectal distention compared to healthy controls [28], which might be suggestive of autonomic dysfunction that leads to disrupted interoceptive signaling, including altered nociception. According to recent interoception definitions [49,50], visceral nociception is considered an interoceptive sub-modality. Visceral hypersensitivity, which refers to abnormal pain signaling to chemical or mechanical stimuli [51], is commonly present in a significant sub-group of patients with IBS, and for this reason, there is a line of research focusing on the molecular mechanisms mediating the transmission and processing of visceral nociceptive signals. Bautzova et al. (2018) [29] showed that 5-oxo-eicosatetraenoic acid (5oxoETE), which is a polyunsaturated fatty acid (PUFA) metabolite, is increased in the colon of patients with IBS with predominant constipation symptoms (IBS-C) and is associated with visceral hyperalgesia. Likewise, a small preliminary randomized controlled trial (RCT) reported that 5-hydroxytryptophan, which enhances serotoninergic transmission, induces rectal hyperalgesia in a sub-group of hypersensitive patients with IBS [43]. In IBS, there is an increased expression of the *transient receptor potential channel subfamily V member 1 (TRPV1)* and *member 3 (TRPV3)* genes, and *TRPV1* mRNA has been associated with increased visceral hypersensitivity [14,21,40,41]. In addition, a double-blind RCT showed that the administration of ebastine, which is an antagonist of the histamine receptor H1 and inhibits the TRPV1 pathway, reduced visceral sensitivity and abdominal pain in patients with IBS [40].

Several studies of patients with IBS have shown structural and functional alterations in brain regions known to be involved in interoceptive processing and in the integration of interoceptive stimuli to emotional regulation. Female patients with IBS presented with a smaller gray matter volume of the insula, the amygdala, and the hippocampus bilaterally, and the left anterior cingulate cortex and larger gray matter volume of the primary and secondary sensorimotor cortex bilaterally compared to healthy women [12,20,31,47]. In addition, the observed changes in the gray matter of specific sub-regions, including the anterior insular cortex, the primary sensorimotor cortex, and the nucleus accumbens, have been significantly associated with rectal pain threshold and the severity of gastrointestinal symptoms independently of the presence of anxiety and depression [12,31,47]. Moreover, lower levels of glutamate + glutamine (Glx) compared to healthy controls were measured in the insula of patients with IBS, which might be implicated in the structural changes of the insular cortices of these patients [12,27]. In contrast, no differences were observed in GABA levels of the anterior insula between patients and controls [27]. Glutaminergic deficiency in the right insula was linked to greater abdominal pain duration, while low glutamate+glutamine concentrations in the left insula were associated with altered cognitive pain regulation [27]. In addition, decreased cortical thickness of the left posterior insula in a large sample of patients with IBS was significantly associated with a specific allele of the *alpha-adrenergic β2 receptor* gene in the presence of a history of sexual abuse [44]. The insula constitutes a crucial hub for interoceptive awareness and its emotional and cognitive regulation [52,53]. In this respect, the aforementioned neurotransmitter level alterations provide evidence of aberrant integration of interoceptive stimuli in IBS, leading to greater emotional arousal and impaired visceral pain modulation. In a similar vein, Gupta et al. (2016) [36] revealed significant inter-correlations between structural changes of the anterior cingulate cortex and stress- and inflammation-related genes. More specifically, they found that in female patients with IBS, there was a reduction in cortical thickness of the left subgenual anterior cingulate cortex, which was associated with a history of severe childhood adversity and specific polymorphisms of the *glucocorticoid receptor NR3C1 gene* and the *interleukin 1β* gene [36]. All these findings suggest that genetic vulnerability and environmental factors (early stress) shape neural pathways that subserve the processing of internal bodily stimuli and contribute to the emergence and regulation of emotional states.

There is a line of studies revealing significant differences in the functional connectivity and activation of the default mode network (amygdala and the dorsal anterior cingulate), the salience network (anterior cingulate cortex and thalamus), the sensorimotor cortex (posterior insula), and the executive control network (dorsal medial prefrontal cortex) in patients with IBS; these connectivity patterns appear to be associated with the presence of visceral hypersensitivity and the severity of gastrointestinal symptoms [32,37,46]. Patients with IBS displayed greater activation of brain regions involved in interoceptive processing and the associated emotional and attentional responses, namely the insula and the anterior cingulate cortex, in conditions of uncertain or anticipated threat (bodily pain) compared to healthy controls and patients with IBD. This finding might suggest that these patients tend to be hypervigilant and emotionally aroused when facing upcoming or ambiguous stimuli originating from their own bodies [37,38]. Another single-blind controlled study showed that there was a different pattern of activation of the brain’s emotional arousal network during the stimulation of interoceptive pathways between patients with IBS and healthy controls [39]. In addition, patients with IBS displayed increased within-network connectivity of the salience network, including the insula and the anterior cingulate, which mediates selective attention to important sensory stimuli and regulates autonomic and behavioral responses [45]. 

Given that IBS is considered a disorder of the gut–brain axis and is commonly associated with psychiatric co-morbidity, mostly depression and anxiety, official guidelines recommend the use of psychological therapies in IBS management, with cognitive behavioral therapy and gut-directed hypnotherapy showing the most solid evidence base [54]. In this context, the involvement of interoceptive processing in IBS pathophysiology has lent growing support to the notion that interoceptive-altering manipulations would be beneficial for patients with IBS, especially in drug-refractory cases. For example, two recent investigations evaluated the effect of two cognitive behavioral therapy (CBT) protocols with interoceptive exposure, one in a group format [15] and the other in a hybrid format [19], on IBS symptom severity and patient’s quality of life and revealed significant improvement. These findings provide promising evidence regarding the usefulness of interoceptive-modulating interventions for patients with IBS. In addition, a recent randomized controlled study has shown that vagus nerve stimulation improved constipation, decreased visceral pain, increased rectal sensation, and reduced the plasma levels of IL-6, TNF-α, and serotonin in patients with IBS with predominant constipation symptoms [23]. Likewise, a small-scale open-label trial and a randomized controlled trial revealed that percutaneous electrical nerve field stimulation (PENFS), which is a vagus nerve modulation intervention, improved gastrointestinal symptoms and reduced abdominal pain in adolescent patients with IBS and other functional abdominal pain syndromes [10,33].

## 4. Other Functional Gastrointestinal Disorders 

Similarly to IBS, the pathophysiology of the other FGIDs has not yet been fully elucidated. However, they are now considered disorders of gut–brain interactions, and visceral hypersensitivity seems to play a crucial part in their pathogenesis [51]. Our search showed that data regarding the role of interoception in other functional gastrointestinal disorders’ pathophysiology are rather sparse. A recent fMRI study in adult patients with Functional Dyspepsia (FD) revealed that they display increased connectivity of the nucleus tractus solitarius (NTS) with regions of the broader interoceptive network, including the insula, the anterior cingulate cortex, and the prefrontal cortex, and a shift of connectivity from the default mode network to the executive control network compared to healthy controls. Furthermore, this connectivity pattern was associated with antral peristaltic frequency [16]. The NTS is the major integration center of visceral input, while the anterior insula is involved in attention to interoceptive stimuli, and these findings suggest that patients with FD display increased attention to distressing gastric interoceptive stimuli and probably employ an altered pattern of cognitive control to cope with interoceptive stimulation [55].

Several structural and fMRI studies have shown distinct patterns of cortical thickness, baseline, and post-stimulation activity, and functional connectivity in areas of the broader interoceptive network, including the dorsal and rostral anterior cingulate cortex and the anterior insula [18,25,30,42] in patients with functional constipation (FC) compared to healthy controls. Furthermore, functional connectivity in these areas was associated with constipation symptoms and the presence of co-morbid anxiety and depression [18]. Likewise, according to a recent study focusing on large-scale intra- and inter-network alterations, patients with functional constipation present with significant functional changes within and across several resting-state neural networks involved in interoceptive processing [17].

Finally, Zucker et al. (2017) [35] assessed the effect of CBT with interoceptive exposure in a group of young children (5–9 years old) with functional abdominal pain and revealed significant reductions in pain, distress about pain, interference caused by pain, and negative affects. These findings are considered preliminary, given that it was a small-scale study with no control group and symptom measurement was based on parents’ ratings. However, they provided promising evidence that teaching children to interpret and cope effectively with visceral bodily signs may contribute to the alleviation of physical and psychological symptoms. 

## 5. Discussion

The current paper detected and critically reviewed original research focusing on the role of interoceptive processing in functional gastrointestinal diseases. The most relevant literature focuses on patients with irritable bowel; however, there are limited investigations, including other patients with FGIDs, such as functional dyspepsia, functional constipation, and functional abdominal pain.

Several investigations have shown that patients with IBS or FD display a heightened visceral sensitivity, decreased visceral pain threshold, and altered interoceptive sensitivity, such as increased attention to unpleasant bodily sensations and difficulties in discriminating between emotions and bodily stimuli [11,13,16,22,24]. At a pathophysiological level, research provides novel evidence that the aforementioned pattern of altered interoceptive processing, including the enhanced signaling of visceral nociceptive stimuli, may be mediated by disturbances to the parasympathetic tone, increased serotoninergic transmission, and increased signaling through the TRPV pathways [14,21,40,41]. Overall, these differences in interoceptive processing may underlie patients with FGIDs amplified emotional and behavioral responses to gut-derived stimuli, which may not only lead to gastrointestinal symptoms exacerbation but may also be associated with the emergence of psychological distress. 

The involvement of interoceptive signaling in FGIDs’ pathophysiology is further supported by a series of neuroimaging studies which reveal distinct patterns of activation and functional connectivity in key components of the interoceptive circuitry, including the insular cortex, the anterior cingulate cortex, the prefrontal cortex, and the somatosensory cortex in patients with FGIDs compared to healthy controls and patients with other chronic gastrointestinal diseases [12,16,17,18,20,25,30,31,32,37,42,45,46,47]. Interestingly enough, structural changes in the insular and the cingulate cortex were associated with reduced levels of Glx without alterations in GABA levels, which suggests that interoceptive impairments may be linked to a reduction in excitatory neurotransmission [12,27]. Moreover, decreased cortical thickness of the left posterior insula and the left anterior cingulate were found to be associated with specific polymorphisms of the alpha-adrenergic β2 receptor, the glucocorticoid receptor, and the *IL-1β* gene, emphasizing the strong cross-communication between interoceptive signaling and endocrine and immune pathways [36,44].

Another important finding of the current review was that the integration of interoception-focused interventions into standard clinical practice might prove promising for patients with FGIDs. A recent well-designed randomized controlled study [15] and two small-scale investigations showed that CBT therapy with Interoceptive Exposure reduced gastrointestinal symptomatology and improved the affective state and quality of life of patients with IBS or FC [19,35]. Interoceptive exposure to visceral sensations helps patients discern pain from other bodily sensations, thus reducing body preoccupations and fear responses to gastrointestinal symptoms and gaining greater emotional awareness and cognitive control over their own internal bodily conditions [35,56,57]. These findings are based overall on a small number of participants, and further larger-scale investigations with a randomized controlled design are needed to corroborate them and determine which patient demographics and clinical characteristics might be associated with more favorable outcomes. In addition, there was a line of preliminary yet promising data suggesting that vagus nerve stimulation may contribute to the attenuation of visceral sensitivity and the improvement of gastrointestinal symptomatology in patients with FGIDs and should be further explored as a treatment option [10,33].

In conclusion, there is a small but constantly growing literature based on a solid methodology (neuroimaging, objective performance tests, and reliable questionnaires) reporting a distinct pattern of interoceptive processing in patients with FGIDs that involves increased interoceptive accuracy and visceral sensitivity, enhanced selective attention to interoceptive cues, disturbed cognitive control, and amplified emotional responses to gastrointestinal sensations. Most of the reviewed studies had small sample sizes and mostly focused on adult female patients with IBS, while males, adolescents, children, elderly, or patients with other FGIDs were either under-represented or completely missing. For this reason, larger investigations, including more sub-populations of patients with FGIDs, should be conducted to corroborate existing evidence and expand their clinical usefulness. Furthermore, interoception-focused psychotherapeutic interventions and vagus nerve stimulation might prove beneficial for patients with drug-refractory diseases and should be considered in FGID clinical management. In line with this, other interoception-manipulating interventions, including breathing exercises and mindfulness approaches, should also be evaluated as potential therapeutic strategies in the field of FGIDs. Finally, research should focus more extensively on the cellular, molecular, and biochemical (neurotransmitter) disturbances observed during interoceptive signaling in patients with FGIDs in order to gain a better understanding of the underlying pathophysiological pathways and design and implement more targeted and effective interventions.

## Figures and Tables

**Figure 1 ijms-25-07633-f001:**
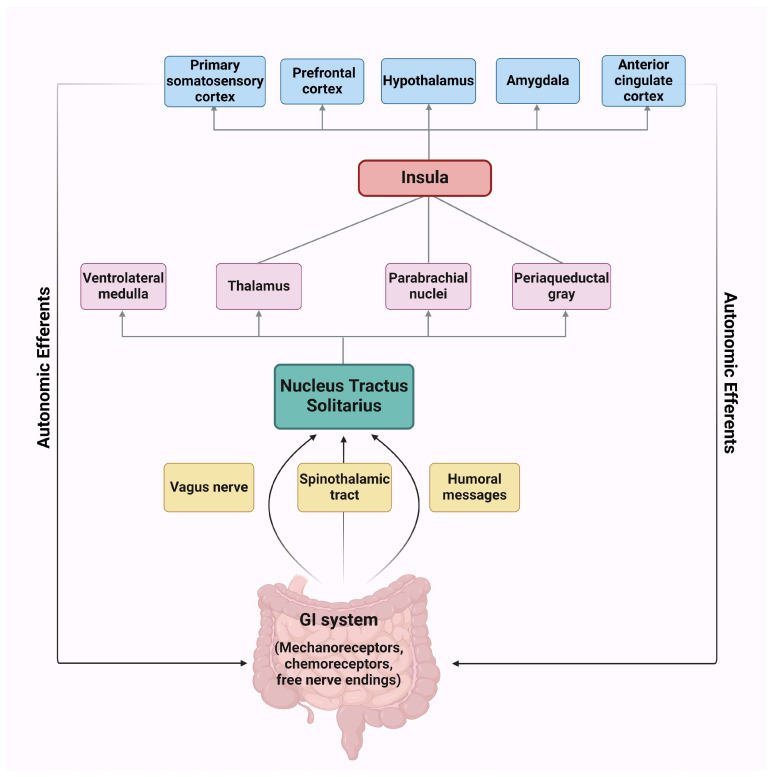
Afferent and efferent interoceptive signaling to the gastrointestinal tract. Created with BioRender.com. Abbreviations: GI, gastrointestinal.

**Table 1 ijms-25-07633-t001:** Search terms used in this review.

Interoception Synonyms and Related Terms	Functional Gastrointestinal Disorders Synonyms and Related Terms
Interoception, interoceptive processing, interoceptive abilities, interoceptive network, nociception, insula, insular cortex, vagus nerve	Functional gastrointestinal disorders, irritable bowel syndrome, functional dyspepsia, functional constipation, functional bloating, functional diarrhea, functional abdominal pain

The search was performed with the use of all possible combinations from the two columns.

**Table 2 ijms-25-07633-t002:** Studies reporting original data on interoceptive signaling in patients with Functional Gastrointestinal Disorders.

Author	Ref	Publication Year	Country	No, Demographic and Clinical Characteristics of Participants	Design	FGID Type	Interoception Assessment
Bora	[10]	2023	USA	20 female patients with IBS aged 11–18 yrs old underwent PENFS	Prospective open-label trial	IBS	
Schulz	[11]	2023	Luxembourg/Germany	12 adult IBS pts/12 matched controls and 20 adult IBD pts/20 matched controls	CS	IBS	Two-Stage Water Load Test
Barazanji	[12]	2022	Sweden	75 female IBS pts/39 controls and 41 female MDD pts/43 controls Age range: 18–65 yrs	CS	IBS	Structural MRI
Bogaerts	[13]	2022	Belgium/Cyprus	38 IBS pts	CS	IBS	ISAQ
Cheng	[14]	2022	China	34 IBS-D/28 healthy controls	CS	IBS	Visual Analogue Scale
Kikuchi	[15]	2022	Japan	154 IBS pts, aged 18–75 (54 under GCBT-IE vs. 60 on WL)	Open-label RCT	IBS	
Sclocco	[16]	2022	USA/S. Korea	15 FD pts/14 controls, age range: 18–65 yrs	CS	FD	fMRI
Zhang	[17]	2022	China	20 FC pts/20 matched controls	CS	FC	fMRI
Duan	[18]	2021	China	41 FCAD/42 FCNAD/43 matched controls	CS	FC	rs-fMRI
Funaba	[19]	2021	Japan	16 IBS pts, 17–65 yrs old, under hybrid CBT-IE	Single-arm, open-label CT	IBS	
Grinsvall	[20]	2021	Sweden/USA	216 female IBS pts aged 18–65/138 controls	CS	IBS	Structural MRI
Grover	[21]	2021	USA/Germany/UK	26 IBS pts/15 healthy controls who received small bowel lipid infusion	Prospective controlled	IBS	Rectal sensitivity test
Mavroudis	[22]	2021	Sweden	36 IBS pts/36 UC pts/14 controls	CS	IBS	Rectal sensitivity test
Shi	[23]	2021	China/USA	42 IBS-C pts randomized to receive either taVNS (21 pts) or sham taVNS (21 pts)	RCT	IBS	
Fournier	[24]	2020	France/Belgium	24 adult IBS pts/18 UC pts/26 controls	CS	IBS	Questionnaires TAS
Hu	[25]	2020	China	29 FC pts/29 healthy controls	CS	FC	Structural MRI
Nan	[26]	2020	China/Australia	IBS pts	CS	IBS	rs-fMRI
Bednarska	[27]	2019	Sweden	39 female IBS pts, aged: 18–57 yrs old/25 controls	CS	IBS	Quantitive MRS
Kano	[28]	2019	Japan/Belgium	27 non-constipated IBS pts/33 healthy controls who underwent colorectal distension	Prospective controlled	IBS	HRV calculation using holter electrocardiography
Bautzova	[29]	2018	France/UK/Italy/Canada	50 IBS pts, aged 20–72 yrs old/20 controls aged 20–76 yrs old	CS	IBS	Ca^2+^ imaging of human sensory neurons
Mugie	[30]	2018	Netherlands	15 adolescent FC pts/10 adolescent FNRFI/15 controls	CS	FC	fMRI
Chua	[31]	2017	Taiwan/Malaysia/USA	29 female IBS pts, 20–50 yrs old/39 controls	CS	IBS	Structural MRI
Icenhour	[32]	2017	Sweden/Germany	41 adult female IBS pts/20 controls	CS	IBS	rs-fMRI
Kovacic	[33]	2017	USA	57 FGID adolescents who received PENFS/47 FGID adolescents who received sham therapy	RCT	Abdominal pain-related FGIDs	
Longarzo	[34]	2017	Italy	19 IBS pts, aged 22–76 yrs old/26 controls	CS	IBS	rs-fMRI
Zucker	[35]	2017	USA	24 FAP pts, 5–9 yrs old, under CBT-IE	Open-label CT	FAP	
Gupta	[36]	2016		73 female patients with IBS/137 healthy controls	CS	IBS	Structural MRI
Hong	[37]	2016	USA	37 IBS pts/37 controls	CS	IBS	Task-dependent fMRI
Huang	[38]	2016	USA	10 IBS pts, aged 16–20 yrs old/10 IBD pts, aged 16–20 yrs old/10 controls, aged 18–22 yrs old	CS	IBS	fMRI
Tanaka	[39]	2016	Japan	16 male IBS pts/16 controls underwent colorectal distension after injection of CRH or saline	Single-blind CT	IBS	PET scan
Wouters	[40]	2016	Belgium/UK/Canada/Netherlands	28 IBS pts received ebastine/27 IBS pts received placebo	Double-blind RCT	IBS	
Zhou	[41]	2016	USA	45 IBS-D pts/40 healthy controls	CS	IBS	Visceral Pain Questionnaire
Zhu	[42]	2016	China	14 FC pts/26 healthy controls	CS	FC	rs-fMRI
Keszthelyi	[43]	2015	Netherlands	15 IBS pts/15 healthy controls who received either 5-hydroxytryptophan or placebo	RCT	IBS	Rectal sensitivity test
Orand	[44]	2015	USA/Germany	121 patients with IBS/209 healthy controls, age range: 18–55 yrs old	CS	IBS	Structural MRI
Gupta	[45]	2014	USA	58 IBS pts/110 controls	CS	IBS	rs-fMRI
Hong	[46]	2014	USA	48 IBS pts/48 controls	CS	IBS	rs-fMRI
Labus	[47]	2014	USA	82 female IBS pts/119 controls, age range: 18–64 yrs old	CS	IBS	Structural MRI

FGID: Functional gastrointestinal disorder; yrs: years; IBS: Irritable Bowel Syndrome; IBS-D: Diarrhea-predominant IBS; IBS-C: Constipation-predominant IBS; IBD: Inflammatory Bowel Disease; FD: Functional Dyspepsia; FC: Functional Constipation; FCAD: Functional Constipation with Anxiety and Depression; FCNAD: Functional Constipation without Anxiety and Depression; FNRFI: Functional Nonretentive Fecal Incontinence; MDD: Major Depressive Disorder; FAP: Functional Abdominal Pain; GCBT-IE: Group Cognitive Behavioral Therapy with Interoceptive Exposure; WL: Waiting List; PENFS: Percutaneous Electrical Nerve Field Stimulation; taVNS: transcutaneous auricular vagal nerve stimulation; CRH: Corticotropin Releasing Hormone; MRI: Magnetic Resonance Imaging; fMRI: Functional MRI; MRS: Magnetic Resonance Spectroscopy; PET: Positron Emission Tomography; CT: Clinical Trial; RCT: Randomized Controlled Trial; CS: Cross-sectional; ISAQ: Interoceptive Sensitivity and Attention Questionnaire; TAS: Toronto Alexithymia Scale.

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
