# Peer review of "Interoceptive Processing in Functional Gastrointestinal Disorders"

_ijms, 2024, doi:10.3390/ijms25147633_

Round 1
Reviewer 1 Report
Comments and Suggestions for Authors
This is a review article well described by the title "Interoceptive processing in functional gastrointestinal disorders", although as the authors admit, most of the data is on IBS.
A total of 38 papers were reviewed, most of which reported functional modalities, such as fMRI. The studies are reviewed in a comprehensive, yet concise way, and logical conclusions are derived from them.
Functional G.I. disorders is an area of much confusion, with many patients not well served by current providers. An understandable review in this area is most needed.
I only have a few minor issues for changes, and are described below.
1. In Table 2 legend, add "CS" for cross-sectional
2. Line 121: Do you mean "augmented" instead of "attenuated"?
3. Line 144: 5-hydroxytryptophan is misspelled.
4. Line 259: "Feelings" is too broad. I think you mean emotion/anxiety-related afferent signaling.
5. Line 266: I doubt that "psychological" illness is secondary to gut signaling. The term is also too broad. It's more likely that the same molecular defects (e.g., channelopathy) that affect the gut also are predisposing towards "mood" disorders by their effects in the brain.
6. Line 273 (also in Results): "Glu-Glx". Do you mean glutamate plus glutamine, or the ratio between the two? Also, glutamine is Gln.
7. Line 276: I assume that "lest" is really "left".
Author Response
Reviewer 1
Comment #1: In table 2 legend, add “CS” for cross-sectional
Author response: Thank you for your comment. It has been done.
Comment #2: Line 121: Do you mean “augmented” instead of “ attenuated”?
Author response: Thank you for pointing this out. It has been replaced (line 129).
Comment #3: Line 144: 5-hydroxytryptophan is misspelled.
Author response: Thank you for your comment. It has been corrected (line 152).
Comment #4: Line 259: “Feelings” is too broad. I think you mean emotion/anxiety-related afferent signaling.
Author response: Thank you for your comment. It has been changed (lines 137 and 268).
Comment #5: Line 266: I doubt that "psychological" illness is secondary to gut signaling. The term is also too broad. It's more likely that the same molecular defects (e.g., channelopathy) that affect the gut also are predisposing towards "mood" disorders by their effects in the brain.
Author response: Thank you for your comment. We rephrased (lines 274-275), given that research findings on the causal associations between interoceptive deficits and psychological symptoms are not yet conclusive.
Comment #6: Line 273 (also in Results): “Glu-Glx”. Do you mean glutamate plus glutamine or the ratio between the two? Also, glutamine is Gln.
Author response: Thank you for your comment. The correct abbreviation is Glx which stands for glutamate plus glutamine. It has been corrected in the manuscript.
Comment #7: Line 276: I assume that “lest” is actually “left”.
Author response: The reviewer is right. It has been corrected (line 286).
Reviewer 2 Report
Comments and Suggestions for Authors
A PDF file with a review is attached.

Comments on the Quality of English LanguageMinor editing of English language required.
Author Response
Reviewer 2
Comment #1: The article lacks the artwork. It will be a valuable addition to this well performed and insightful research. It is required to present the following graphics:
- A graphical abstract.
- A scheme of afferent and efferent interoceptive processing in the gastrointestinal tract.
- A summary of FGIDs.
- Causative effective interoception FGIDs relation.
Author response: Thank you for your comment. The submission was already accompanied by a graphical abstract. A scheme of afferent and efferent interoceptive processing in the gastrointestinal tract has been added
Comment #2: List of abbreviation is missing. There are a lot of non explained acronyms and abbreviations QoL) that should be listed in an acronym definition manner.
Author response: Indeed, the reviewer is right. It has been done.
Comment #3: A linguistic layer and an article structure are well done. Only a slight English proofreading and editing corrections are suggested.
Author response: Thank you for your suggestion. The manuscript has been proofread and edited by a colleague who is fluent in English.
Comment #4:
Introduction:
- It is suggested to add more information and details about the FGIDs so to keep the proportion between descriptions of the FGIDs and the interoception.
- The Authors presented the negative” inclusion criteria for their review. What were the positive” inclusion criteria, if any? There is a lot of other studies on interception FGIDs, including inflammatory bowel diseases, Crohn s disease, ulcerative colitis, and microbiota gut brain axis diseases see references below, point 7. in this review) that must be included in this review.
- Table 1 is ordered according to the date of original article publication. In my opinion, it would be more informative and scientifically supported to arrange these studies according to the research design or the FGID type.
- What are the limitations of this review?
Author response:
- Thank you for your comment. It has been done (lines 31-34 and line 37)
- At the end of the introduction section, it is clearly stated which are the inclusion criteria, namely studies written in English, published between 2014-2024 and focusing on interoceptive processing in human subjects with a diagnosis of FGIDs. IBD are not considered functional gastrointestinal diseases and for this reason they were not included in the current review
Comment #5:
Studies characteristics:
- This chapter is extremely informative and detailed, yet chaotic. Perhaphs, it would be better to structurize this chapter into separated paragraphs according to the colums in Table 1? Moreover, it is suggested to add few introductory sentences for each table descriptor, e g.., types of designs, types of FGIDs, types pf neuroimaging, types of clinical trial human studies.
Author response: Thank you for your suggestion. However, we included table 2 in order to present in a comprehensive, organized and reader-friendly way all necessary information regarding the characteristics of the reviewed studies.
Comment #6:
Discussion
- It is suggested to provide more details on on going and future therapies of interoceptive processing associated FGIDs. Please, refer to
- Psychiatry Clin Neurosci. 2023 Jul 8; 77 10): 530 540
- https doi org 10 1016 j tins 2020 09 009
Author response: Thank you for your comment. The second article you suggested (Bonaz et al, 2021) was already in our reference list (ref.4). We also added useful information derived from the other review in our manuscript and included it in our references.
Comment #7. To enrich their valuable review, the Authors should discuss the following research:
- Inflammatory bowel disease:
- https doi org 10 3389 fpsyt 2021 680878
- https doi org 10 1093 ecco jcc jjab076 204
iii. https doi org 10 1038 s41380 024 02612 7
- https doi org 10 3389 fpsyt 2022 833423
- https doi org 10 20524 aog 2023 0813
- Crohn s disease
- https doi org 10 1111 nmo 13593
- https doi org 10 1111 nmo 12844
iii. https doi org 10 1002 brb3 2003
- https dx doi org 10 21037 qim
- Ulcerative colitis
- https doi org 10 3389 fpsyt 2020 00229
- https doi org 10 3389 fpsyg 2021 671493
iii. https doi org 10 1093 ecco jcc jjad080
- Microbiota gut brain axis disorders
- Major depressive disorder
- https doi org 10 3389 fpsyt 2023 1273439
- https doi org 10 1016 j jpsychires 2022 10 020
- https doi org 10 1186 s12888 023 05168 y
- https doi org 10 1038 s41398 024 02907 x
- Schizophrenia
- https doi org 10 1093 schizbullopen sgaa067
- https doi org 10 1016 j eclinm 2024 102673
- https doi org 10 9758 cpn 2023 21 2 252
iii. Alzheimer s disease
- https doi org 10 1016 j tins 2022 08 005
- https doi org 10 1016 j biopsych 2022 02 955
- https doi org 10 1523 JNEUROSCI 2578 20 2021
- https doi org 10 1016 j cortex 2023 02 009
Autism spectrum disorder
- https doi org 10 1007 s10803 019 04079 w
- https doi org 10 1002 aur 1880
- https doi org 10 1186 s13229 016 0104 x
- https doi org 10 1002 aur 2289
Author response: Thank you for these comprehensive literature suggestions. However, the scope of this review was to focus only on FGIDs and this is the reason we did not include studies on IBD, neurodevelopmental, neurodegenerative and other disorders.
Round 2
Reviewer 2 Report
Comments and Suggestions for Authors
Accepted.
Comments on the Quality of English LanguageMinor editing of English language required.